# Hyponatremia—Long-Term Prognostic Factor for Nonfatal Pulmonary Embolism

**DOI:** 10.3390/diagnostics11020214

**Published:** 2021-02-01

**Authors:** Anca Ouatu, Madalina Stefana Mihai, Daniela Maria Tanase, Cristina Gena Dascalu, Nicoleta Dima, Lacramioara Ionela Serban, Ciprian Rezus, Mariana Floria

**Affiliations:** 1Faculty of General Medicine, “Grigore T. Popa” University of Medicine and Pharmacy, 16 University Street, 700115 Iasi, Romania; ank_mihailescu@yahoo.com (A.O.); nicoleta2006r@yahoo.com (N.D.); ionela.serban@umfiasi.ro (L.I.S.); ciprianrezus@yahoo.com (C.R.); floria_mariana@yahoo.com (M.F.); 2Department of Internal Medicine, IIIrd Medical Clinic, “Sf. Spiridon” Emergency Hospital, 1 Independentei Street, 700111 Iasi, Romania; 3Department of Gastroenterology, Iași County “Sf. Spiridon” Emergency Hospital, 700111 Iași, Romania; madalina_mihai@yahoo.com; 4Department of Medical Informatics and Biostatistics, “Grigore T. Popa” University of Medicine and Pharmacy, 700115 Iasi, Romania; cdascalu_info@yahoo.com; 5Department of Physiology, “Grigore T. Popa” University of Medicine and Pharmacy, 700115 Iasi, Romania; 6Department of Cardiology, Emergency Military Clinical Hospital, 7-9 General Henri Mathias Berthelot Street, 700483 Iasi, Romania

**Keywords:** hyponatremia, pulmonary embolism (PE), prognostic factor

## Abstract

Over recent years, studies have shown that in patients with left-sided heart failure, arterial hypertension, and acute coronary syndrome, hyponatremia is a negative prognostic factor. In this context, there is raising interest in the association between hyponatremia and pulmonary embolism (PE). This retrospective cohort study includes 404 consecutive patients with confirmed acute nonfatal pulmonary embolism divided into four groups according to their sodium fluctuation pattern. The primary outcome was all-cause mortality and determining the recurrence rate among patients with nonfatal PE using serum sodium levels as a continuous variable. Patients with acquired and persistent hyponatremia had a significantly higher rate of mortality rate than those in the normonatremia group (12.8% and 40.4%, OR- 7.206, CI: 2.383–21.791, *p* = 0.000 and OR-33.250, CI: 11.521–95.960, *p* = 0.000 vs. 2%, *p* < 0.001, respectively). Mean survival time decreases from 23.624 months (95% CI: (23.295–23.953)) in the normonatremia group to 16.426 months (95% CI: (13.17–19.134)) in the persistent hyponatremia group, statistically significant (*p* = 0.000). The mean survival time for all patients was 22.441 months (95% CI: (21.930–22.951)). The highest recurrence rate was recorded at 12 and 24 months in the acquired hyponatremia group (16.7% and 14.1%, respectively). Serum sodium determination is a simple and cost-effective approach in evaluating the short and long-term prognosis in patients with acute PE.

## 1. Introduction

Hyponatremia is the most frequent electrolyte disorder in hospitalized patients (~30%). In-hospital mortality is increased by over 50% in patients with hyponatremia compared with those with normonatremia [1]. Moreover, not only in-hospital mortality but also long-term mortality (1 to 5 years) is higher in patients with persistent (6.2%) and acquired hyponatremia (5.9%), compared with those with corrected hyponatremia (3.9%); the lowest mortality rate is recorded in normonatremic patients (1.8%) [2]. The same study reveals that hyponatremic patients have more comorbidities compared with patients with normal sodium levels. Maintaining normal serum sodium levels is crucial for proper cellular activity. Sodium level reflects the intracellular fluid volume but tells us nothing about total body sodium or extracellular fluid volume; its role in modulating the immune cell function is also a matter of interest in the last years [3].

The negative prognostic value of low sodium levels is associated with various conditions like acute myocardial infarction [3,4], congestive heart failure [5,6,7,8] and neoplastic syndromes [9,10,11]. The classic mechanisms of hyponatremia like the activation of the renin–angiotensin–aldosterone system and elevated catecholamine levels in heart failure leading to vasoconstriction and decreased glomerular filtration rate are well known. Additionally, hyponatremia is a predictor of poor prognosis in chronic renal disease and pulmonary arterial hypertension [12,13]; also, patients with chronic hyponatremia suffer from neurological damage in the absence of cerebral edema [14]. The prognostic significance of hyponatremia in chronic left heart failure (LHF) reflects the strong correlation between serum sodium and plasma neurohormone concentrations, such as norepinephrine, renin, and angiotensin II, all of which are linked to poor outcome in advanced LHF. Neurohormone-mediated, nonosmotic release of vasopressin accounts for the fall in sodium levels in these patients. There are also strong correlations between hyponatremia and right ventricular failure, suggesting that low serum sodium is a global marker of circulatory maladaptation in heart failure, regardless of whether left or right ventricular function is compromised. [13].

Approximately 20–40% of patients with acute nonfatal pulmonary embolism (PE) make up 5–21% of all-cause mortality within the first 90 days [15]. Hyponatremia is common in patients with PE [16,17,18] while hypernatremia is more frequently associated with venous thromboembolism [16,19]. Long-term as well as short-term prognostics for PE have not been studied, although mortality 4 years after an acute PE episode was ~35% [20]. While hypo- and hypernatremia are both associated with an increased risk of venous thromboembolism, morbidity, and mortality; only hyponatremia is associated with hospital readmission [16].

The question arises whether sodium levels may influence the clinical course and prognosis of patients with pulmonary embolism. According to the latest guidelines, risk stratification in nonfatal PE is based on biological and radiologic markers of right ventricular (RV) dysfunction and myocardial injury markers assessing the severity of PE [21]. Pulmonary Embolism Severity Index (PESI) is a very useful and effective tool for short-term mortality prediction. A recent study compared PESI and troponin as predictors for identifying patients with low 30-day mortality risk, with clearly better results in favor of PESI [22]. There are also data demonstrating the utility of using PESI for long-term prognosis in patients with PE. Dentali et al. [23] demonstrated the accuracy of PESI in predicting the mortality risk at 6 months and 1 year. Unlike troponin levels, PESI determination identified more precisely the patients at lower risk for a negative outcome. However, assessing the cardiac status alone should not be considered a full-prognostic factor in patients with PE; other factors are also important, namely age, pulmonary status, and renal function. In fact, in a previous study, we focused on the importance of renal function in patients with PE and its impact on long-term prognosis [24].

Sodium level determination is a common practice for all in-hospital patients. Hyponatremia is a well-known marker of neurohormonal activation in patients with left-sided heart failure and is highly correlated with plasm norepinephrine, renin, and angiotensin II levels [25], being also a marker of right ventricular dysfunction [13]. In this context, arises the hypothesis according to which hyponatremia, as a single prognostic factor or in association with other prognostic factors, may predict poor outcome in patients with acute nonfatal PE.

## 2. Materials and Methods

This retrospective cohort study was conducted in a university-affiliated tertiary-referral medical center and the medical data used has been previously described [24]. In summary, consecutive patients with confirmed acute nonfatal PE, without any previous oral anticoagulation record/oral coagulation-naive were identified retrospectively over a 4-year period.

Confirmed PE was diagnosed according to the published guidelines: computed tomography pulmonary angiography (CTPA) positive for PE, ventilation-perfusion lung scintigraphy with a high probability of pulmonary thromboembolism or ventilation-perfusion lung scintigraphy with an intermediate probability associated with the presence of distal deep vein thrombosis on compression ultrasonography. Recurrent PE was defined as PE developed after successful acute treatment, meeting the same imaging criteria mentioned above for confirmed PE.

Transthoracic echocardiography was performed within the first 24 h of admission in all study patients using a SonoScape SSI-8000 ultrasound, linear transducer with the frequency of 3–5.1 MHz. Parasternal short-axis view, apical 4 chamber view, and subxiphoid window were used to assess the right ventricular (RV) function. RV dysfunction was diagnosed if tricuspid annular plane systolic excursion (TAPSE) < 22 mm, if the acceleration time of pulmonary ejection was < 80 ms, if the pulmonary artery systolic pressure was > 30 mmHg (using the maximal systolic velocity of the tricuspid regurgitation jet and the diameter of the inferior vena cava measured during expiration) and if the RV telediastolic diameter was > 35 mm.

Information on the patient’s profile (age, sex, medical history, comorbidities, the imaging modality used to diagnose PE, echocardiographic parameters, and blood test results) was obtained from their medical records.

Normal serum sodium levels were considered 135–145 mEq/l. Hyponatremia was defined as a serum sodium level < 135 mEq/L. Serum sodium levels were determined on admission (baseline level) and on the following pre-specified time points after admission: 6, 12, and 24 months. If a patient was readmitted to hospital (for recurrent PE or any other condition requiring hospitalization in the internal medical clinic) in-between the pre-specified time points, biochemical determinations were assigned to the closest above-mentioned time point.

All study patients were aged > 18 and gave written informed consent.

The primary outcome was all-cause mortality and recurrence rate among patients with nonfatal PE using serum sodium levels as a continuous variable. Our secondary outcome was the prognostic strength of the PESI score by adding serum sodium levels.

Patients with fatal pulmonary thromboembolism who were associated with shock or blood pressure (BP) < 90 mmHg (according to the ESC guidelines of diagnosis and treatment on acute pulmonary embolism) and patients enrolled in other studies were excluded from this study.

### Statistical Analysis 

Continuous data were presented as mean value ± standard deviation. Categorical data were presented as percentages. To assess the relationship between categorical variables crosstabulation and chi-square tests were performed. Kaplan–Meier analysis was used to assess survival rates between compared groups and Mantel–Cox log-rank was used to test for statistical significance. Odds-ratio was used to estimate the risk associated with hyponatremia among deceased patients and the correlation between PESI and serum sodium levels.

The last hospital discharge form or prescription drug order was considered as confirmation of patient survival status. Hemodynamically unstable patients (with shock or hypotension) were considered a high-risk category according to published guidelines and were excluded from the study.

Non-high-risk patients were divided into intermediate or low-risk groups based on their troponin and/or proBNP levels and/or RV dysfunction or signs of myocardial damage. In patients with PE the severity of PE was estimated by PESI, a prognostic score for in-hospital or 30-day outcome, in order to achieve the secondary outcome of the study.

This study aims to evaluate any variation in serum sodium level during the first admission, and in the medium-term at specified time points for nonfatal PE. In this sense the cohort was divided into 4 groups: group 1—normonatremia (sodium > 135 mmol/L from admission to discharge), group 2—corrected hyponatremia (baseline sodium < 135 mmol/L but then normalized during hospital stay), group 3—acquired hyponatremia (baseline sodium > 135 mmol/L, then decreased to < 135 mmol/L) and group 4—persistent hyponatremia (sodium < 135 mmol/L from admission to discharge).

For all statistical analyses, a 95% confidence interval was assumed. A *p*-value of less than 0.05 was considered statistically significant and a value < 0.01 was considered highly statistically significant. Statistical analysis was performed using Statistical Package for the Social Sciences (SPSS) version 24.0.

Also investigated was the influence of the detected comorbid conditions associated with hyponatremia and high PESI score on mortality and recurrence risks. We used a forward stepwise binary logistic regression model; the model’s quality of the model was evaluated using the Hosmer and Lemeshow goodness-of-fit test. Adjusted odds ratios (AOR) and their 95% confidence intervals (95% CI) were estimated. For all statistical tests, association was considered statistically significant at *p* < 0.05.

## 3. Results

Of the 438 patients that met the criteria for acute PE, 34 patients (7.76%) were excluded from the study because they were lost to follow-up. The cohort study included 404 patients, 205 male (50.7%) and 199 females (49.3%) with a median age of 62.32 ± 14.267 years. Sodium levels were determined at admission (baseline level), and again at 6, 12 and 24 months (1559 serum sodium titrations).

### 3.1. Baseline Characteristics

According to their sodium fluctuation pattern, patients were divided into four groups: group 1 (normonatremia, n = 250, 61.9%),group 2 (corrected hyponatremia, n = 29, 72%),group 3 (acquired hyponatremia, n = 78, 19.3%),group 4 (persistent hyponatremia, n = 47, 11.6%).

Baseline characteristics of the study cohort are presented in Table 1.

The most frequent comorbidities, in decreasing order, were: chronic cardiovascular diseases (coronary artery disease 53.7%, heart failure 52%, arterial hypertension 42.3%), metabolic disorders (dyslipidemia 24.5%, diabetes mellitus 18.6%), and chronic obstructive pulmonary disease (24.5%). Chronic cardiac disease status was also reflected by the percentage of patients with RV dysfunction assessed echocardiographically, the highest rate associated with statistically significance being recorded in group 3 (acquired hyponatremia, 73 patients, 93.6%, *p* = 0.022). Additionally, group 3 and 4 patients were older; median age being 65.27 ± 14.03 and 64.85 ± 14.31 months, respectively. The analysis of the collected data revealed that most patients with high-risk PESI score were in the acquired hyponatremia group (76.9%).

### 3.2. Long-Term Mortality

Overall, 35 (8,66%) of the 404 patients died during the monitoring period/follow-up. Patients with acquired and persistent hyponatremia had a significantly higher mortality rate than patients in the normonatremia group (12.8% and 40.4%, OR- 7.206, CI:2.383–21.791, *p* = 0.000 and OR-33.250, CI: 11.521–95.960, *p* = 0.000, respectively) (Table 2 and Table 3).

Figure 1 shows the Kaplan–Meier survival curve of the study cohort stratified into 4 groups according to sodium levels.

Six-month survival rate showed a significat decrease and 2-year mortality rate remained high among patients with acquired and persistent hyponatremia (group 3 and 4) compared with normonatremic patients. Long-term survival at 2 years was similar in group 1 (normonatremic patients) and group 2 (corrected hyponatremia). The mean survival time decreased from 23.624 months (95% CI: (23.295–23.953)) in the normonatremic group to 16.426 months (95% CI: (13.717–19.134)) in the persistent hyponatremia group, statistically significant (*p* = 0.000). The mean survival time for all patients was 22,441 months (95% CI: (21.930–22.951)).

In our study the highest 6-month mortality rate was 29.8% and was recorded in group 4 compared to 35.7% in Ng’s study [20]. Of the 35 patients, 24 died within the first 6 months, most of them belonging to groups 3 and 4. At 12 months the highest mortality rate was identified in group 4, similar to the data reported by Ng et al. (40.4% versus 41.1%).

In the patients with hyponatremia and PE, there was an 11.855-fold increase in overall mortality risk compared to patients with normal sodium levels. In group 2, mortality risk is not very high (OR-1.750, CI: 0.197–15.517, *p* = 0.611), but it is significant in group 4 (OR-33.250, CI: 11.521–95.960, *p* = 0.0001) (Table 3).

### 3.3. Serum Sodium Level and PESI as Early Outcome Predictors

The PESI score is used to identify patients at high risk of 30-day mortality, but some data claim it is useful to evaluate medium-term prognosis.

In the study group, patients with high PESI scores were at 3.598-fold higher increased risk of early death. Adding hyponatremia as a prognostic factor in all patients with high PESI score resulted in higher mortality risk (OR = 9.043, CI: (3.357–24.359) *p* = 0.000) but lower than when hyponatremia was used as a single prognostic factor (OR = 11.855, CI: (4.489–31.306) *p* = 0.000) in all patients. The addition of high PESI score to hyponatremia in the four study groups showed a higher mortality risk in all hyponatremia groups (group 2—OR = 1.524, CI: (0.170–13.680), *p* = 0.705; group 3—OR = 4.923, CI: (1.543–15.709), *p* = 0.003; group 4—OR = 28.632, CI: (9.486–86.423), *p* = 0.0001) but a slightly decreased mortality risk compared to the cases in which hyponatremia was independently investigated (Table 4). However, the association of these two parameters showed a significantly higher mortality risk in groups 3 and 4 (acquired and persistent hyponatremia).

### 3.4. Long-Term Recurrence

Recurrence rate showed an increase in groups 3 and 4 (52 patients, 66.7%; 21 patients, 44.7% respectively), with the highest percentage being recorded at 12 months in all groups (Table 4). At 6 months, the highest recurrence rate was recorded in group 2 (corrected hyponatremia). Overall, patients with acquired hyponatremia (group 3) had the highest recurrence rate of PE over the study period (Table 5). In fact, a decrease in serum sodium levels during the first year after an episode of acute PE is associated with a 16.7% risk of recurrence and remains high, at 14.1%, at 2 years. Group 2 patients, with correction of hyponatremia at 24 months, had a recurrence rate of 10.3% in the first year but at 2 years no recurrence episode was recorded (Figure 2).

### 3.5. Other Comorbidities

As our patients had multiple comorbidities, several combinations of independent variables were tested; the investigated comorbidities were chronic obstructive pulmonary disease, arterial hypertension, chronic coronary syndrome, diabetes mellitus, dyslipidemia and chronic heart failure, included in the model as independent categorical variables.

Overall, in the entire study population, the significant risk factors for death are the dyslipidemia (OR = 12.113, CI: (1.563 ÷ 93.862), *p*= 0.017) and the decrease in sodium level (OR = 8.023, CI: (3.706 ÷ 17.370), *p* = 0.000); diabetes mellitus and high PESI score are borderline (OR = 0.409, CI: (0.163 ÷ 1.023), *p* = 0.056 and OR = 3.000, CI: (0.995 ÷ 9.043), *p* = 0.051, respectively). In our study, chronic obstructive pulmonary disease, arterial hypertension, chronic coronary syndrome and chronic heart failure had no influence over mortality in our study. No significant risk factors for PE recurrence were identified.

In the particular cases of patients with corrected hyponatremia or acquired hyponatremia compared with the patients with normonatremia, the binary logistic regression did not identify significant predictors between the existing comorbidities.

In the patients with persistent hyponatremia compared to normonatremia patients, the single identified significant independent predictor was again the low sodium level; the presence of the other comorbidities did not influence mortality (Table 6).

## 4. Discussion

In our study groups, the distribution by sex (50.7% women vs. 49.3% men) and associated significant comorbidities (mainly cardiovascular disease, the most frequent being heart failure and arterial hypertension) was approximately equal; 86.6% of patients had echocardiographically confirmed RV dysfunction or elevated serum troponin/BNP levels. RV dysfunction was most common in the group of patients with acquired hyponatremia (93.6%), slightly higher compared to the group with persistent hyponatremia (91.5%) and significantly higher than in patients with normonatremia (83.2%). This suggests that the occurrence or persistence of hyponatremia represents a marker of adverse evolution in patients with RV dysfunction. Forfia et al. [13] have shown that low serum sodium levels are a global marker of circulatory maladaptation in patients with heart failure, regardless of whether the left or right ventricular function is compromised. Patients with hyponatremia also had significantly more advanced RV systolic dysfunction confirmed by echocardiography, including marked differences in tricuspid annular plane systolic excursion between the hyponatremia and normonatremia groups [13]. Approximately 50% of the patients with chronic heart failure with reduced ejection fraction and persistent hyponatremia at discharge had the worst clinical outcomes with a 20% mortality rate [26]. 

In 2010, Scherz et al. [17] were the first to report that firstly found hyponatremia as a frequent feature and at the same time an independent predictor of short-term mortality and hospital readmission in PE patients. In the following years, similar findings were also reported but in smaller cohorts [18,19,21,27,28]. A recent study has shown that hyponatremia is associated with adverse hemodynamics and reduced survival in patients with intermediate-risk PE, more specifically demonstrating negative correlations with the echocardiographic markers of RV dysfunction as mean PAP and right atrial pressure [28].

In terms of PE mortality risk, on admission, 95.8% of patients were identified as being at intermediate risk and 4.2% being at low risk. The mean age of the patients was over 60 years, which explains the heavy share of the associated comorbidities. There were 35 deaths, all from the group of patients at intermediate risk.

In 2010, Scherz et al. [17] demonstrated the short-term prognostic value (30 days) of hyponatremia in patients with acute symptomatic PE. One of the strengths of the study was the great number of studied PE patients (13.728) with a prevalence of hyponatremia of 21%. In our study, the prevalence of hyponatremia was 38%, mainly because of the associated comorbidities. The same study shows that hyponatremia is associated with an increased risk of short-term death (30 days) and an increased risk of readmission compared to those with normal sodium levels.

In 2016, a meta-analysis by Zhou et al. revealed the poor short-term prognosis in patients with acute PE [29]. It is important to mention that the studies included in this meta-analysis were using different designs. Scherz et al. stratified their cohort based on three serum sodium levels (<130, 130–135, >135 mmol/L) and correlated them with 30-day mortality (the main goal of the study) and the five PESI severity classes. Similarly, Tamizifar et al. correlated serum sodium level (severe hyponatremia <135, mild hyponatremia 135–137) with 30-day mortality and sPESI subclasses but the sample size was smaller [18]. None of these studies aimed to extend the follow-up beyond 30 days. Korkmaz et al. Additionally studied the relationship between sodium level on admission and 30-day PE-related mortality [27].

One of the limitations of the study conducted by Scherz et al. was that the authors did not establish the prognostic value of persistent or acquired hyponatremia. Given the longer follow-up period, we aimed at evaluating the clinical significance of changes in sodium level and variations in biological values depending on water intake, medication and acute diseases. In our patients, persistent hyponatremia at 2 years is associated with a higher risk of mortality (40.4%) compared with 2% in normonatremic patients. Also, our patients with persistent hyponatremia at the time of diagnosis and at 2 years follow-up had a 33-fold higher risk of death than patients with PE and normal sodium levels (OR 33.250, CI: 11.521–95.960, *p* = 0.0001). In 2013, Ng et al. have shown that sodium fluctuations after acute PE predict acute and long-term outcomes [30]. Thus, persistent hyponatremia increased the risk by 5.6-fold (95% CI 2.08–15.0, *p* = 0.001). The elevated percentage of risk in our study is probably due to the presence of comorbid disease, the most common being ischemic heart disease and chronic kidney disease, both of them having a strong influence on sodium homeostasis. In contrast with the previous study, the prevalence of ischemic heart disease was higher (51% versus 15%) because of the profile of the patients admitted to our clinic-elderly with multiple comorbidities (mean age 61.06 ± 14.30 years), the situation being similar for chronic kidney disease (32% versus 6%). Indeed, advanced age is most likely to be accompanied by atherosclerotic and thromboembolic heart diseases, often patients associating heart, pulmonary or renal diseases with thromboembolic phenomena.

The goal of this study was also to demonstrate that patients should be analyzed globally because in the short or long-term various diseases become entangled. Comparing the 4 groups, the highest mortality rate and also the highest prevalence of ischemic, chronic renal or chronic lung disease (COPD) were found in group 4 (persistent hyponatremia). In contrast, the percentage of patients with dyslipidemia was lower in group 4 compared to group 3 (20.5% versus 12.8%), probably explained by the presence of possible liver dysfunction associated with the other comorbidities. Additionally, diabetes was more frequent in group 3 compared to group 4. Thirty-five of the 404 patients died during the monitoring period, most of them in groups 3 and 4 (10 patients in group 3–12.8% and 19 patients in group 4–14.4%, *p* = 0.0001). Analyzing survival curves, we observed that the correction of hyponatremia caused similar mortality as in the normonatremic patients, whereas the occurrence or persistence of hyponatremia led to an increase in mortality rate in patients with PE.

The mortality rate of 8.66% found in our study was much lower than that of 38.8% reported by Ng et al., but in their study, the duration of follow-up was longer (up to 5 years) compared to our study (up to 2 years). Also, the 2-year mortality rate in the group with persistent hyponatremia was similar with 1-year mortality rate (40.4% versus 41.1%), the risk of mortality increasing with each year (58.9% at 3 years and 66.1% at 5 years) [30].

Most of the deaths occurred within the first 12 months after the onset of PE (Table 2), and especially in the first 6 months in patients with persistent hyponatremia. In our study, the 6 month-mortality rate was 29.8% in group 4 in our study (mostly from patients with acquired or persistent hyponatremia), compared to 35.7% in the same group in the study of Ng et al. Thus, it is confirmed that the persistence of uncorrected hyponatremia is associated with increased 1-year mortality, especially in the first 6 months.

PESI is the most used predictive score in acute PE at 30 days. Additionally, recent studies have demonstrated that this score can also predict long-term mortality [23]. This score includes several clinical and laboratory parameters. Thus, the question arises whether adding serum sodium levels to the PESI score will improve the prognostic accuracy. In 2013, Dentali and Riva demonstrated that adding serum sodium levels to the PESI score results in a slight improvement in the short-term adverse prognostic identification [17]. In our study, we first found an increased mortality risk when adding the high PESI score to hyponatremia in all patients (OR = 9.043, CI: (3.357–24.359) *p* = 0.000), especially in patients with acquired and persistent hyponatremia (OR = 4.923, CI: (1.543–15.709) *p* = 0.003; OR = 28.632, Cl: (9.486–86.423) *p* = 0.000, respectively). However, the risk identified by the combination of these two factors was lower than the hyponatremia taken separately in the same groups 3 and 4 (OR-7.206, CI: 2.383–21.791, *p* = 0.000, and OR-33.250, CI: 11.521–95.960, *p* = 0.000, respectively). This proves the predictive power of hyponatremia as a short and long-term negative prognostic factor in patients with acute PE, and also that the addition of serum sodium level to PESI score does not improve the accuracy of this score in terms of long-term mortality in PE. Another study, similarly found that incorporating the sPESI in the risk analysis did not alter the significance of hyponatremia on the study outcomes (*p* = 0.74, *p* = 0.09, *p* = 0.04 respectively, for the corrected, acquired, and persistent hyponatremia) [30].

Regarding the long-term recurrence of PE, there are other well-known risk factors such as age, sex, location of thrombosis, or D-dimer level [31]. In our study group, there is a high recurrence rate mainly due to therapeutic noncompliance. We wondered whether the level of hyponatremia correlates with the recurrence the same way the mortality rate does. The occurrence of hyponatremia within the first 12 months after PE correlates with the highest recurrence rate (16.7%), which is higher than the recurrence rate of 12.8% in the group with persistent hyponatremia. Contrariwise, correction of hyponatremia (group 2) is associated with the absence of recurrences at 2 years (Figure 2). In our study, the highest recurrence rate was recorded at 1 year in both patients with normonatremia and in those with corrected, acquired, or persistent hyponatremia. It is worth mentioning that hyponatremia and not its persistence is associated with a higher recurrence rate. We interpreted these results in the possible context of the overlapped new diseases that influence the vital prognosis.

Our study patients presented several comorbidities, especially heart diseases (arterial hypertension, chronic coronary syndrome and chronic heart failure), and metabolic diseases (diabetes mellitus, dyslipidemia). Of all of these, only dyslipidemia proved to be a marker of mortality (OR = 12.113, CI: (1.563 ÷ 93.862), *p*= 0.017) in all study patients and none of them of PE recurrence. When we studied separately for the corrected, acquired and persistent hyponatremia, the binary logistic regression did not identify any comorbidity as a significant predictor. According to a study, the prevalence of atherosclerosis is significantly higher in patients with spontaneous venous thromboembolism (VTE) than in patients with VTE secondary to known risk factors and controls [32].

In our study, 18.56% of patients had diabetes mellitus and 24.50% had dyslipidemia. Observational studies have demonstrated an association between dyslipidemia and VTE. [33,34] Dyslipidemia is a well-established risk factor for atherothrombotic disorders. Besides their strong effects on atherogenesis, lipids and lipoproteins could influence hemostasis by modulating the expression and function of procoagulant, fibrinolytic, and rheological factors [35]. Triglycerides, for example, seem to increase factor VII levels, plasminogen activator inhibitor (PAI-1) levels, and blood viscosity. LDL promotes platelet activation and tissue factor expression. HDL has anti-atherothrombotic properties that may result from the inhibition of platelet aggregation, reduction of viscosity, suppression of tissue factor activity, and PAI-1 activity levels, and enhancement of inactivation of factor Va by activated protein C [36]. Because of these possible biological effects on the hemostatic system, lipids may also contribute to the development of venous thrombosis [33]. This study demonstrated that in men older than 50 years with idiopathic deep vein thrombosis, hypercholesterolaemia is an independent risk factor with an OR of 2.6, in line with Kawasaki’s studies (Kawasaki et al. 1995, 1997), suggesting that hypercholesterolaemia seems to play a role in the pathogenesis of thromboembolic events [34].

Diabetes mellitus is known to be an inflammatory disease that may cause dyslipidemia (which is associated with hypercoagulation, endothelial dysfunction and increased platelet aggregation), hypertension and abnormal coagulation [33]. We found diabetes mellitus as a borderline mortality risk factor for the entire study group (OR = 0.409, CI: (0.163 ÷ 1.023), *p* = 0.056) with no influence on mortality or recurrence in the three hyponatremia groups. Another very recent study reported similar results regarding the association between VTE, diabetes mellitus, and obesity (*p* = 0.051 and 0.061, respectively) [37].

The relationship between cardiovascular risk factors and PE without identifiable risk factors is very relevant. At present, PE is classified as unprovoked in the absence of risk factors such as neoplasm, pregnancy, trauma, surgery, immobilization, or some diseases. If the relationship between cardiovascular risk factors and PE is demonstrated, this would lead to the development of new strategies for the prevention of PE [37].

Our study has several strengths. A significant number of patients were studied over a fairly long period of time, most of them (except for those who died) undergoing 3 subsequent clinical and laboratory assessments, sodium level determination included. The presence of cardiac comorbidities is both a weakness and a strength because the incidence and prevalence of cardiovascular diseases are constantly increasing in the developed countries and we rarely diagnose a single disease in an elderly patient. Therefore, the association of PE with cardiovascular disease will increase as the incidence of PE increases. The clinical course of hyponatremia in patients with acute PE over a longer period has been less studied so far; it is also more complex because it implies more interactions between serum sodium fluctuations, overlapping medical conditions (acute or chronic), and associated medication (diuretics, anticoagulants). Our study confirms the prognostic power of hyponatremia and suggests that adding the serum sodium levels to the PESI score does not improve the prognostic accuracy in patients with PE. Given the smaller sample size, and in order to avoid bias as much as possible, in our study the follow-up period was extended beyond 30 days, but limited to 24 months.

### Study Limitations

Our study also had some limitations. Over 50% of the study patients had heart diseases, of which left-sided heart failure may interfere with the presence of hyponatremia and RV dysfunction. Additionally, our hospital is a regional tertiary center, so the investigated patients are more likely to present complex disorders, with a high mortality risk that may influence the end-results. The older age of patients is also a factor that can influence short or long-term mortality. We could not make more precise correlations between sodium levels and PE recurrence, because the recurrence occurrence time could not be accurately identified. We do not have any information on the administration of diuretics in these patients and the presence or absence of other diseases that could influence sodium such as cirrhosis or nephrotic syndrome. Additionally, iatrogenic dilutional hyponatremia could not be excluded since most of the study patients had comorbidities that required fluid resuscitation and adjustment of diuretics.

As to the anticoagulant treatment at home, according to the international guidelines, most patients were on treatment with VKA (vitamin K antagonists) for 3–6 months or indefinitely depending on their particularities. At 6 months, 91.7% of the deceased patients received no anticoagulant treatment or took the medication incorrectly; 40.3% of those within recurrent PE did not follow the treatment as prescribed. At 12 months, 66.7% of the deceased patients were not on anticoagulant treatment or took the medication incorrectly, as did 63.9% of those with recurrent EP. No deaths were recorded at 2 years; of those diagnosed with recurrence, they maintained the same percentage as at 12 months. Since most of the deceased patients and of those diagnosed with or recurrence had acquired or persistent hyponatremia, we believe that the anticoagulant treatment failure could have influenced morbidity and mortality of our study patients, thus affecting the final outcomes. On the other hand, some studies claim that the duration of anticoagulation does not influence morbidity and mortality after a first PE episode [38].

## 5. Conclusions

In conclusion, we can say that hyponatremia is an independent, long-term (2 years), negative prognostic factor in patients with PE, prognosis that worsens as hyponatremia persists longer or, at worst, becomes permanent. Serum sodium determination is a simple method that can be widely used, being helpful in establishing the subsequent treatment for PE.

## Figures and Tables

**Figure 1 diagnostics-11-00214-f001:**
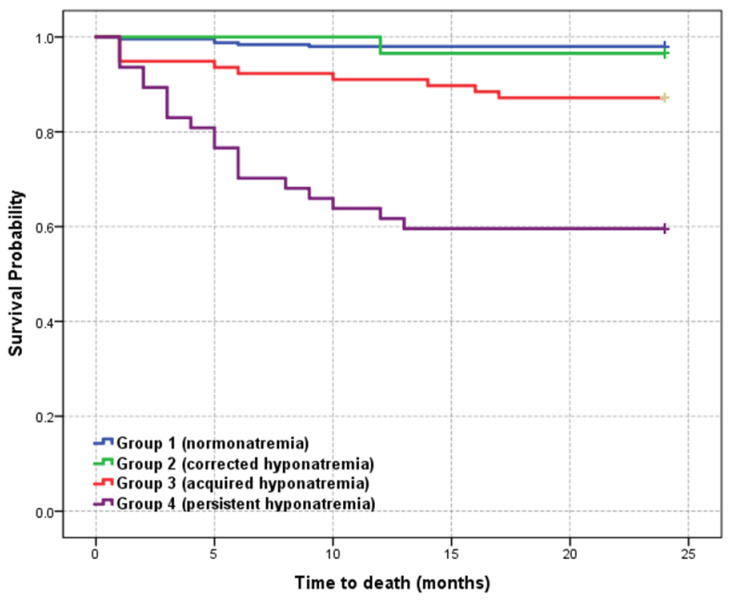
Kaplan–Meier survival curve of the study cohort according to serum sodium levels.

**Figure 2 diagnostics-11-00214-f002:**
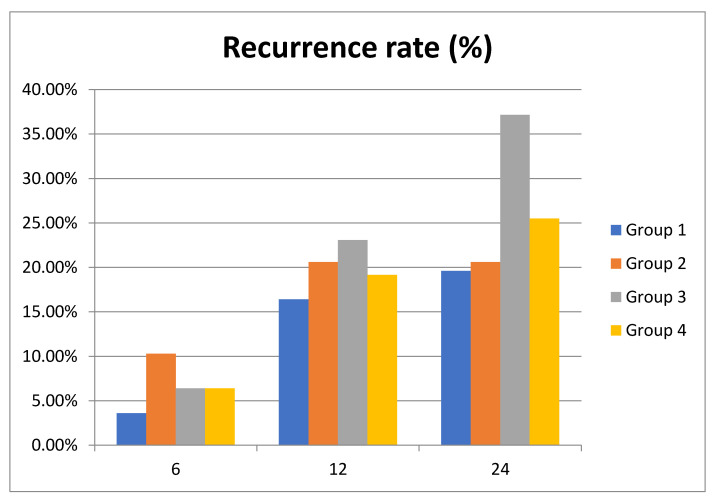
Recurrence rate of pulmonary embolism (PE) in the study cohort at 6, 12 and 24 months.

**Table 1 diagnostics-11-00214-t001:** Baseline Characteristics of the Study Cohort.

Parameters	Group 1 (Normonatremia)	Group 2 (Corrected Hyponatremia)	Group 3 (Acquired Hyponatremia)	Group 4 (Persistent Hyponatremia)
	*N* = 250	*N* = 29	*N* = 78	*N* = 47
Females–no. (%)	131 (52.4)	15 (51.7)	32 (41.0)	21 (44.7)
Males–no. (%)	119 (47.6)	14 (48.3)	46 (59.0)	26 (55.3)
		*p* = 0.945	*p* = 0.079	*p* = 0.331
Mean age (± SD)- years	61.06 ± 14.30	61.14 ± 13.51 *p* = 0.979	65.27 ± 14.03 * *p* = 0.023	64.85 ± 14.31 *p* = 0.097
Chronic obstructive pulmonary disease–no. (%)	58 (23.2)	3 (10.3) *p* = 0.113	21 (26.9) *p* = 0.502	17 (36.2) *p* = 0.060
Arterial hypertension–no.(%)	110 (44.0)	10 (34.5) *p* = 0.327	37 (47.4) *p* = 0.594	14 (29.8) *p* = 0.070
Chronic coronary disease–no. (%)	129 (51.6)	14 (48.3) *p* = 0.735	46 (59.0) *p* = 0.254	28 (59.6) *p* = 0.315
Diabetes mellitus–no. (%)	48 (19.2)	5 (17.2) *p* = 0.799	17 (21.8) *p* = 0.616	5 (10.6) *p* = 0.160
Chronic kidney disease	80(32%)	6(20,6) *p*= 0.226	22(28,2) *p*= 0.534	25(53,3) *p*= 0.412
Dyslipidemia–no. (%)	73 (29.2)	4 (13.8) *p* = 0.079	16 (20.5) *p* = 0.132	6 (12.8) * *p* = 0.019
Chronic heart failure–no. (%)	132 (52.8)	15 (51.7) *p* = 0.913	41 (52.6) *p* = 0.971	23 (48.9) *p* = 0.627
Right ventricular dysfunction (echocardiographically)–no. (%)	208 (83.2)	26 (89.7) *p* = 0.371	73 (93.6) * *p* = 0.022	43 (91.5) *p* = 0.150

Note: * *p* < 0.05 Group 3 vs. Group 1.

**Table 2 diagnostics-11-00214-t002:** Mortality rate in patients with pulmonary embolism (PE).

All-Cause Mortality–no. (%)	Group1	Group 2	Group 3	Group 4
	*N* = 250	*N* = 29	*N* = 78	*N* = 47
At 6 months	4 (1.6%)	-	6 (7.6%)	14 (29.8%)
At 12 months	5 (2.0%)	1 (3.4%)	8 (10.2%)	19 (40,4%)
At 24 months	5 (2.0%)	1 (3.4%)	10 (12.8%)	19 (40.4%)
Total	5 (2.0%)	1 (3.4%)	10 (12.8%)	19 (40.4%)

**Table 3 diagnostics-11-00214-t003:** Association between hyponatremia and mortality in patients with pulmonary embolism (PE).

	Mortality Rate	OR	95% Confidence Interval	*p*
All patients (404)	35/369(8.7)	11.855	4.489–31.306	0.0001
Group 1 (normonatremia)	5/250(2)	-	-	-
Group 2 (corrected hyponatremia)	1/29(3.4)	1.750	0.197–15.517	0.611
Group 3 (acquired hyponatremia)	10/78(12.8)	15.959	2.383–21.791	0.0001
Group 4 (persistent hyponatremia)	19/47(40.4)	33.250	11.521–95.960	0.0001

**Table 4 diagnostics-11-00214-t004:** Association between hyponatremia and high Pulmonary Embolism Severity Index (PESI) regarding mortality in patients with pulmonary embolism (PE).

	Mortality Rate	OR	95% Confidence Interval	*p*
All patients (404) + PESI	26/118(22.03%)	9.043	3.357–24.359	0.000
Group 1 (normonatremia)	0	-	-	-
Group 2 (corrected hyponatremia) + PESI	1/22(4.54%)	1.524	0.170–13.680	0.705
Group 3 (acquired hyponatremia) + PESI	8/60(13.33%)	4.923	1.543–15.709	0.003
Group 4 (persistent hyponatremia) + PESI	17/36(47.22%)	28.632	9.486–86.423	0.000

**Table 5 diagnostics-11-00214-t005:** Number of patients with PE recurrence.

	Group 1	Group 2	Group 3	Group 4	Total
	*N* = 250	*N* = 29	*N* = 78	*N* = 47	*N* = 404
At 6 months	9 (3.6%)	3 (10.3%)	5 (6.4%)	3 (6.4%)	20 (5.0%)
At 12 months	32 (12.8%)	3 (10.3%)	13 (16.7%)	6 (12.8%)	54 (13.4%)
At 24 months	8 (3.2%)	0	11 (14.1%)	3 (6.4%)	22 (5.4%)

**Table 6 diagnostics-11-00214-t006:** Results of binary logistic regression analysis for different combinations of predictors and clinical outcome–patients with persistent hyponatremia vs. patients with normonatremia.

Significant Predictors	OR	95%CI	*p*-Value
Independent variables: Comorbidities + Hyponatremia/Dependent variable: Death			
Persistent hyponatremia	33.250	11.521 ÷ 95.960	0.000
Independent variables: Comorbidities + Hyponatremia + PESI/Dependent variable: Death			
Persistent hyponatremia	32.930	11.192 ÷ 96.892	0.000
High PESI score	5.629	1.172 ÷ 27.029	0.031

## Data Availability

Data is contained within the article.

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
