# Peer review of "Hyponatremia—Long-Term Prognostic Factor for Nonfatal Pulmonary Embolism"

_diagnostics, 2021, doi:10.3390/diagnostics11020214_

Round 1

Reviewer 1 Report

The manuscript is well written, the results are nicely presented. However, the concerns with the manuscript are regarding novelty of results and conclusions as the idea of using hyponatremia as a prognosis marker in PE patients has been extensively discussed. In fact a meta-analysis by Zhou et al (DOI 10.1016/j.ijcard.2016.11.120 ) that included 8 studies showed that the 30 days mortality rate for hyponatremia PE patients is as around 15.9, compared to 7.4 for normonatremia PE patients, suggesting hyponatremia could indicate poor short-term prognosis in patients with PE and could be a useful marker of mortality.

As most papers are presenting 30-dayes mortality rates, the merit of this manuscript would be the long-term follow-up of patients and I suggest this should be better highlighted. Also, in the introduction and discussion, the presentation of the state-of-the-art of using hyponatremia as a mortality/prognosis marker in PE patients should be better presented as there are many articles published on this subject. The results of the current manuscript should be better compared to the already existing results on this subject.

Author Response

Response to Reviewer 1 Comments

Point 1: The manuscript is well written, the results are nicely presented. However, the concerns with the manuscript are rearding novelty of results and conclusions as the idea of using hyponatremia as a prognosis marker in PE patients has been extensively discussed. In fact a meta-analysis by Zhou et al (DOI 10.1016/j.ijcard.2016.11.120 ) that included 8 studies showed that the 30 days mortality rate for hyponatremia PE patients is as around 15.9, compared to 7.4 for normonatremia PE patients, suggesting hyponatremia could indicate poor short-term prognosis in patients with PE and could be a useful marker of mortality.

As most papers are presenting 30-dayes mortality rates, the merit of this manuscript would be the long-term follow-up of patients and I suggest this should be better highlighted. Also, in the introduction and discussion, the presentation of the state-of-the-art of using hyponatremia as a mortality/prognosis marker in PE patients should be better presented as there are many articles published on this subject. The results of the current manuscript should be better compared to the already existing results on this subject.

Response 1: We thank you very much for your appreciation and are grateful for your recommendations You are right. Hyponatremia as a long-term prognosis marker in PE patients was our goal ,also. We improved the introduction and the discussion as requested.

We didn’t find any new data regarding hyponatremia and longterm mortality in PE except Ng’ s article in 2013.(Ng AC, Chow V. Fluctuation of serum sodium and its impact on short and long-term mortality following acute pulmonary embolism. PloS one. 2013; Apr 19;8(4):e61966.), so we compared with that.

We have introduced the review by Ng as reference 20.

We also compared Zhou et al with our data in the discussion section.

“In 2016, a meta-analysis by Zhou et al. revealed the poor short-term prognosis in patients with acute PE [29]. It is important to mention that the studies included in this meta-analysis were using different designs. Scherz et al. stratified their cohort based on three serum sodium levels (<130, 130-135, >135 mmol/L) and correlated them with 30-day mortality (main goal of the study) and the five PESI severity classes. Similarly, Tamizifar et al. also correlated serum sodium level (severe hyponatremia< 135, mild hyponatremia 135-137) with 30-day mortality and sPESI subclasses, but the sample size was smaller. [18] None of these studies aimed to extend the follow-up beyond 30 days. Korkmaz et al. also studied the relationship between sodium level on admission and 30-day PE-related mortality. [27]”

We improved our Kaplan-Meier analysis by adding more accurate data regarding the time of death for the non-survivor patients.

The mean survival time decreased from 23.624 months (95% CI: (23.295-23.953)) in the normonatremic group to 16.426 months (95% CI: (13.717-19. 134)) in the persistent hyponatremia group, statistically significant (p = 0.000). The mean survival time for all patients was 22.441 months (95% CI : (21.930-22.951)).

Compared with Ng’s study , in our study the highest 6-month mortality rate was 29.8% in group 4 versus 35.7% in Ng’s study. [20]

Introduction:

“Sodium concentration reflects the intracellular fluid volume but tells us nothing about total body sodium or extracellular fluid volume [3].

The prognostic significance of hyponatremia in chronic left heart failure (LHF) reflects the strong correlation between serum sodium and plasma neurohormone concentrations, such as norepinephrine, renin, and angiotensin II, all of which are linked to poor outcome in advanced LHF. Neurohormone-mediated, nonosmotic release of vasopressin accounts for the fall in sodium concentration in these patients. In fact, there are also strong correlations between hyponatremia and right ventricular failure, suggesting that low serum sodium is a global marker of circulatory maladaptation in heart failure, regardless of whether left or right ventricular function is compromised [13].

Discussion:

“In 2010, it was demonstrated the short-term prognostic value (30 days) of hyponatremia in patients with acute symptomatic PE. [17] One of the strengths of the study was the large number of the studied PE patients (13.728) with a prevalence of hyponatremia of 21%. In our study, the prevalence of hyponatremia was 38%, mainly because of the associated comorbidities. The same study shows that hyponatremia is associated with an increased risk of short-term death (30 days) and an increased risk of readmission compared to those with normal sodium levels.

In 2016, a meta-analysis by Zhou et al. revealed the poor short-term prognosis in patients with acute PE [29]. It is important to mention that the studies included in this meta-analysis were using different designs. Scherz et al. stratified their cohort based on three serum sodium levels (<130, 130-135, >135 mmol/L) and correlated them with 30-day mortality (main goal of the study) and the five PESI severity classes. Similarly, Tamizifar et al. also correlated serum sodium level (severe hyponatremia< 135, mild hyponatremia 135-137) with 30-day mortality and sPESI subclasses, but the sample size was smaller. [18] None of these studies aimed to extend the follow-up beyond 30 days. Korkmaz et al. also studied the relationship between sodium level on admission and 30-day PE-related mortality. [27]

The mortality rate of 8.66% found in our study was much lower than that of 38.8% reported by Ng et al., but the duration of follow-up was longer (up to 5 years) compared to our study (up to 2 years). Also, the 2-year mortality rate in the group with persistent hyponatremia was similar with 1-year mortality rate (40.4% versus 41.1%), the risk of mortality increasing with each year (58.9% at 3years and 66.1% at 5 years) [30].

 In our study, we first found an increased mortality risk when adding the high PESI score to hyponatremia in all patients (OR = 9.043, CI: (3.357 – 24.359) p=0.000), especially in patients with acquired and persistent hyponatremia (OR = 4.923, CI: (1.543 – 15.709) p = 0.003; OR = 28.632, CI: (9.486 – 86.423) p = 0.000, respectively). However, the risk identified by the combination of these two factors was lower than by hyponatremia alone in the same groups 3 and 4 (OR- 7.206, CI: 2.383-21.791, p=0.000, and OR-33.250, CI: 11.521-95.960, p=0.000, respectively). This proves the predictive power of hyponatremia as a short and long-term negative prognostic factor in patients with acute PE, and also that the addition of serum sodium level to PESI score does not improve the accuracy of this score in terms of long-term mortality in PE. Another study, similarly found that incorporating the sPESI in the risk analysis did not alter the significance of hyponatremia on the study outcomes (p=0.74, p=0.09, p=0.04, respectively, for the corrected, acquired, and persistent hyponatremia) [30].

Point 2: Extensive editing of English language and style required

Response 2: We edited the text once more. We appreciate your suggestions and are highly grateful for your help in improving the quality of this manuscript.

Reviewer 2 Report

The manuscript presents several typos. Please see the attached pdf with the manual annotations.

The authors claim on lines 172-178 that the median survival rate is of several months. This is methodologically wrong. Survival rate is measured in percentages at a given time point. Also, it is absolutely wrong to state that the median survival rate is of 23.664 months, since such number depends on the observation period. The authors should report the rates in percentages and hold on reporting average survival time. 

The Kaplan-Meier analysis is performed using only four time points (0,6, 12 and 24 months). Please use the exitus date of the patients to regenerate such analysis. Also, use a higher-quality graph with the right labels on the y axis.

The authors discuss about the non-improvement of risk mortality prediction when using PESI plus sodium levels with respect to using only serum levels. This should be backed up with data. 

Finally, this reviewer believes stronger logistic regression models using comorbidities as covariates could be of interest given the data the authors have gathered. 

Round 2

Reviewer 1 Report

The manuscript in general is improved, there are however a few aspects to be corrected:

line 23 - no comma after "hyponatremia"

line 37 - "in patients with.." not "is patients..."

line 48 - I would suggest to use "sodium levels" in plural when referring to the sodium blood concentrations

line 65 - the first time "pulmonary embolism" is mentioned in the manuscript, PE should be mentioned in brackets.

line 75 - PESI (meaning?)

line 87 - plasmatic levels (plural) of...

line 90 -  "arises" is misplaced

line 174 - echocardiography

line 180 - the percentage should be near 35 not 404.

line 244 - Baseline characteristics of the study cohort

Table 3 and 4 - need formatting

Author Response

  1. Extensive editing of English language and style required

Thank you for this suggestion. We improved the text extensively, regarding English language and style, as you will see in the manuscript attached.

  1. The manuscript in general is improved, there are however a few aspects to be corrected:

line 23 - no comma after "hyponatremia"  - Done

line 37 - "in patients with.." not "is patients..." – Done

line 48 - I would suggest to use "sodium levels" in plural when referring to the sodium blood concentrations - Done

line 65 - the first time "pulmonary embolism" is mentioned in the manuscript, PE should be mentioned in brackets. - Done

line 75 - PESI (meaning?) - Done

line 87 - plasmatic levels (plural) of... - Done

line 90 -  "arises" is misplaced - Done

line 174 – echocardiography - Done

line 180 - the percentage should be near 35 not 404. - Done

line 244 - Baseline characteristics of the study cohort - Done

We made all the changes required. We thank you  very much for your appreciation and are grateful for your recommendations  as it is hard to always find all the typos.

Table 3 and 4 - need formatting

We adjusted the format of the tables as required.

We appreciate your suggestions and are highly grateful for your help in improving the quality of this manuscript.
